# Phytochemical Profile and Antimicrobial Potential of Extracts Obtained from *Thymus marschallianus* Willd

**DOI:** 10.3390/molecules24173101

**Published:** 2019-08-26

**Authors:** Mihaela Niculae, Daniela Hanganu, Ilioara Oniga, Daniela Benedec, Irina Ielciu, Radu Giupana, Carmen Dana Sandru, Nina Ciocârlan, Marina Spinu

**Affiliations:** 1Department of Clinical Sciences, Division of Infectious Diseases, University of Agricultural Sciences and Veterinary Medicine, 400374 Cluj-Napoca, Romania; 2Department of Pharmacognosy, “Iuliu Haţieganu” University of Medicine and Pharmacy, 400010 Cluj-Napoca, Romania; 3Department of Pharmaceutical Botany, “Iuliu Haţieganu” University of Medicine and Pharmacy, 400010 Cluj-Napoca, Romania; 4Botanical Garden of Academy of Sciences, 2002 Chișinău, Moldova

**Keywords:** *T. marschallianus*, polyphenols, antibacterial

## Abstract

*Thymus marschallianus* Willd. is a Lamiaceae species spread in a large variety of habitats worldwide. The aim of the present research was to analyse two different samples belonging to this species, one obtained from the spontaneous flora and one from culture. The total polyphenols, flavonoids, and phenolic acid contents were spectrophotometrically determined. Qualitative and quantitative analysis of polyphenols was performed by an *HPLC-DAD-ESI (+)-MS* method. For the antibacterial assay, the well-diffusion and the broth microdilution methods were used. Analysis of polyphenols revealed for both samples the presence of flavonoids like luteolin, quercetin, apigenin and their derivatives, but also of rosmarinic acid and methyl-rosmarinate. Differences regarding the amount of these compounds were emphasized. Significantly larger amounts of flavonoids were found for the sample harvested in the spontaneous flora, while for the rosmarinic acid, larger amounts were found for the cultured sample. Both samples displayed promising antibacterial activity, particularly towards Gram positive organisms. *T. marschallianus* represents, therefore, a rich source of polyphenolic compounds that prove its promising potential as a medicinal species.

## 1. Introduction

The Lamiaceae family is one of the largest plant families worldwide, comprising more than 700 species, included in over 230 genera, which can be found in a large number of habitats. It is one of the flowering plant families that presents the largest number of species with medicinal properties [1,2].

Among the species of this family, *Thymus* genus is one of the most important and taxonomically complex genera, comprising over 300 species worldwide, distributed from Europe to Asia, North Africa, and the Canary Islands [3], the vast majority of these species being found in the Mediterranean areas and in the surroundings [4]. In Europe, this genus comprises 66 species that can be found in arid, temperate, and cold regions [5]. It is one of the genera that has created a lot of taxonomy confusions among botanists worldwide and especially in the European continent, due to high population variability, even though data about morphology and chemical composition exist [6]. Difficulties in taxonomical interpretation regard especially the section *Serpyllum*, which appears to be the oldest and the one that is the richest in number of species, comprising more than 150 species [7].

Traditional medicine reports the use of the *Thymus* species for their expectorant, anti-inflammatory and antimicrobial properties [8]. Scientific evidence on the antibacterial activities of some *Thymus* sp [8,9,10] are reported to be due both to their composition in essential oils [1,11] and also to their composition in phenolic compounds [10].

*Thymus marschallianus* Willd. or *T. pannonicus* All., known as the Hungarian thyme, is an Eurasian species which has created over time a lot of taxonomy problems, as the two taxa are considered distinct species in different areas [12]. Similarities and differences can be found from morphological and chemical points of view. The morphological differences between the two species are related to the presence of trichomes on the leaves surface, as *T. pannonicus* has leaves covered with long multi-cellular trichomes, while *T. marschallianus* has glabrous leaves. From a chemical point of view, researches on molecular markers did not show a clear differentiation between the two taxa [7]. This is in agreement with the European [5] and Romanian Floras [13,14,15], where *T. pannonicus* and *T. marschallianus* are considered synonymous. However, in the Flora of Serbia [16,17], *T. pannonicus* All. and *T. marschallianus* Willd. are cited as distinct species. Data concerning analysis of volatile compounds have been reported for samples belonging to Serbian species and support this fact [4]. Differences between the two species regard the chemical composition of the essential oil. The lemon-scented populations of *T. pannonicus* contain more than 70% of cis- and trans-citral (neral and geranial), the major components of the essential oil [18]. On the other side, the populations of *T. marschallianus* Willd. contain large amounts of thymol and carvacrol [4,19]. This brings additional arguments which support the fact that these species represent different taxa.

The aim of the present work was to provide scientific evidence on the fact that *T. marschalianus* is an important and valuable source of polyphenolic compounds, which can prove its significant potential in the development of natural antibacterial agents. The originality of this study consists in the fact that it is the first report on the presence of polyphenolic compounds and on their antimicrobial activity and, at the same time, it is the first study that provides the comparison between two types of samples, harvested from culture and from the spontaneous flora. To the best of our knowledge, there are no reported studies on the *T. marschallianus* polyphenolic composition and its biological effects.

## 2. Results and Discussion

### 2.1. Analysis of Polyphenolic Compounds by High-Performance Liquid Chromatography-Diode Array Detection-Electro-Spray Ionization Mass Spectrometry (HPLC-DAD-ESI (+)-MS)

A *HPLC-DAD-ESI (+)-MS* method was carried out for the qualitative and quantitative determination of the polyphenolic compounds from *T. marschallianus* samples harvested from culture (TMc) and in spontaneous flora (TMs) [20,21,22]. The method allows the simultaneous analysis of different polyphenols by a single pass through the analytical column. Separation of all examined compounds was performed in 25 min. Concentrations of identified polyphenolic compounds in both analyzed samples are shown in Table 1, in the order of their retention time and the HPLC-DAD chromatograms of these samples are shown in Figure 1. Quantitative determination was performed using the external standard method. Analysis revealed for both samples the presence of luteolin, quercetin, apigenin and their derivatives, but also of methyl-rosmarinate and rosmarinic acid, as important compounds in the composition of this species.

In each sample, nine polyphenolic compounds were identified: rosmarinic acid and its derivative, methyl-rosmarinate, four flavonoidic glycosides of quercetin, luteolin, and apigenin: luteolin-7-*O*-glucuronide, quercetin-3-*O*-glucoside, quercetin-7-*O*-arabinoside, apigenin-7-*O*-glucuronide and three flavonoidic aglycones: quercetin, luteolin, and apigenin. Results were expressed as mg compound/g vegetal material (d.w) and revealed quantitative differences between the identified compounds. In both samples, the highest amounts of polyphenolic compounds were obtained for luteolin-7-*O*-glucuronide (20.20 ± 0.85 mg/g d.w. in TMc and 38.63 ± 0.92 mg/g d.w. in TMs respectively) and methyl-rosmarinate (29.39 ± 0.97 mg/g d.w. in TMc and 24.68 ± 0.85 mg/g d.w. in TMs respectively). These two compounds were therefore identified as being the majority compounds in both samples. The highest concentration of luteolin-7-*O*-glucuronide was obtained for TMs (*p* < 0.001), while the amount found for methyl-rosmarinate was higher (*p* < 0.01) in TMc. It can also be clearly noticed that not only its derivative, methyl-rosmarinate, but also rosmarinic acid was biosynthesized in significantly larger amounts (*p* < 0.001) in the sample obtained from culture (Table 1). These compounds, together with the other compounds identified in the composition of this species may represent important markers in the chemical composition of this species. Presence of rosmarinic acid and luteolin-7-*O*-glucuronide was previously reported for other species of *Thymus*, as *T. sibtorpii*, *T. serpyllum*, *T. praecox* ssp. *arcticus*, T. *austriacus*, *T. dacicus* or *T. pulegioides* [10,23]. Presence of these compounds was also found in the composition of *T. pannonicus* [24,25], which appear as a chemical similarity between the two species. Differences between the two species seem to appear for the methyl-rosmarinate, the methylated derivative of rosmarinic acid, the present study being the first report in the composition of a *Thymus* species and also in the composition of *T. marschallianus*.

Statistically significant differences (*p* < 0.001) were observed when comparing concentrations obtained for the other identified compounds that are found in lesser amounts in these samples. TMs presented the most elevated content for the three identified aglycones (quercetin, luteolin, and apigenin) and their derivatives: quercetin-3-*O*-glucoside (isoquercitrin), luteolin-7-*O*-glucuronide and apigenin-7-*O*-glucuronide. The only exception was noticed for quercetin-7-*O*-arabinoside concentration that proved to be higher in TMc (Table 1).

Presence of these compounds appear as an important feature of the chemical composition of the species belonging to the *Thymus* genus, as previously reported in scientific literature [10,23], but, at the same time, the present study appears to be the first one to report the presence of the quercetin aglycone and its derivatives, of the apigenin aglycone and its derivative and of the aglycone luteolin in the chemical composition of *T. marschallianus.* The data obtained in this study can, therefore, represent a starting point for more detailed studies that can help in identifying the chemical composition of the two *Thymus* species that are involved in the taxonomic confusions.

Under ex situ conditions, monitoring of this species during the vegetation period has shown a good adaptation to growth conditions, with higher sizes of morphological parameters and a higher green mass yield compared to spontaneous flora. The plants were multiplied by fragments of rhizomes. Being transplanted under conditions similar to those in the natural populations, the plants developed well and flourished abundantly from the first year of vegetation. Plants undertake the complete ontogenetic cycle, demonstrating a high potential for adaptation in culture conditions.

Differences between tested samples regarding especially the amount of rosmarinic acid and its derivative (methyl-rosmarinate), can be therefore explained to these pedoclimatic and agrotechnical conditions to which these plants are adapted, which are better controlled in culture. 

The content of rosmarinic acid in *T. marschallianus* was comparable to that found by other species of Lamiaceae. The results obtained for the cultured sample were superior to those reported for *Hyssopus officinalis, Salvia officinalis, Rosmarinus officinalis, Melissa officinalis, Mentha piperita, Satureja hortensis, Thymus vulgaris* and *Ocimum basilicum* (1.33-7.84 mg rosmarinic acid/g), but lower than in *Origanum vulgare* (12.40 mg rosmarinic acid/g). The presence of methyl rosmarinate alongside the rosmarinic acid makes this species a rich source of rosmarinic acid [2,26,27]. 

The other compounds identified in the composition of these species, as the two quercetin derivatives (quercetin-3-*O*-glucoside and quercetin-7-*O*-arabinoside), the apigenin derivative (apigenin-7-*O*-glucuronide) and the three aglycones (quercetin, luteolin, and apigenin) are reported for the first time.

### 2.2. The Content of Total Polyphenols, Flavonoids, and Phenolic Acid

Besides the identified polyphenolic compounds, there are many others in the extracts which play an important role for the antimicrobial activities of this species. Thus, the comparative spectrometrical quantification of total polyphenols, flavonoids, and phenolic acids contents were performed (Table 2).

The total polyphenolic content (TPC) of these extracts was estimated using the Folin–Ciocâlteu method and the obtained results were expressed as mg of gallic acid equivalents (GAE)/g of dry plant material (d.w). Determination of total flavonoids content (TFC) was carried out using the aluminium chloride colorimetric method and the results were expressed as mg of rutin equivalents (RE)/g d.w. The content of phenolic acids (TPA) was expressed as mg of rosmarinic acid equivalents (RAE)/g d.w. [28,29,30,31].

Large amounts of total polyphenols, flavonoids, and phenolic acids were determined for both samples. No statistically significant differences between the tested samples were noticed, except for the content of flavonoids which appears to be higher (*p* < 0.001) in the spontaneous flora sample (28.98 ± 0.32mg RE/g for TMs and 16.69 ± 0.51mg RE/g for TMc). 

The obtained amounts for this species are superior to the ones obtained from other Lamiaceae species as *Ocimum basilicum* [30] or *Rosmarinus officinalis* [32]. *Origanum vulgare* and *Mentha* sp. showed superior results for TPC than *T. marschallianus*, but inferior for TFC and TPA [29,33].

### 2.3. Antibacterial Assays

The results obtained for the in vitro antimicrobial activity evaluation are presented in Table 3 and Table 4.

Both extracts presented in vitro inhibitory activity against some of the tested microorganisms. Selected bacterial strains displayed different in vitro susceptibility towards the samples. The most intense effect was recorded towards three Gram positive bacteria (*Staphylococcus aureus* > *Bacillus cereus* > *Staphylococcus pseudintermedius*), with inhibition zones between 16–22 mm diameter, while *Enterococcus faecalis* showed susceptibility only against TMs. The lowest sensibility was noticed in case of Gram-negative bacteria, with the smallest values of the inhibition zone (*p* < 0.001) compared to Gram positive and also the standard antibacterial agent, gentamicin (Table 3). None of the extracts was able to inhibit *Salmonella typhimurium* growth, while a reduced inhibitory action (*p* < 0.001) was exhibited by both TMc and TMs towards *Salmonella enteritidis* in comparison to gentamicin (10 ± 0.00 mm, 10.33±0.58 mm and 18 ± 0.00 mm, respectively).

Based on the obtained values of the inhibition zone diameter, the antimicrobial potential was considered more intense in case of TMs, since compared to gentamicin this sample exhibited increased (*Staphylococcus aureus*, *Staphylococcus pseudintermedius*) (*p* < 0.01) or similar (*Bacillus cereus*) efficacy (*p* > 0.05) (Table 3). 

Values obtained for MICs and MBCs (Table 4) indicated a higher antimicrobial efficacy of the TMs extract and also a bactericidal activity (MIC index < 4), against all tested bacteria except for *Staphylococcus aureus*. The bactericidal effect was observed for TMs extract against two Gram positive, *Staphylococcus pseudintermedius* and *Bacillus cereus.* If connecting the results obtained in the chemical composition of this species, as flavonoids appear to be the compounds that are majority in the TMs extract, it is suggested that these compounds may be responsible for the antimicrobial activity of this species, harvested from the spontaneous flora or may significantly contribute to this biological activity. This aspect is supported also by the Pearson coefficients indicating a strong positive correlation between the total flavonoids content and the antimicrobial potential evaluated towards *Staphylococcus aureus* (*r^2^* = 0.977), *Staphylococcus pseudintermedius (r^2^* = 0.944), and *Bacillus cereus* (*r^2^* = 0.989).

Despite the fact that considerable progress had been made in developing and implementing viable strategies aimed to prevent and control antibiotic resistance, an augmenting incidence of multi-drug resistant bacterial strains has been reported and the consequences of multidrug organisms’ emergence and spread continue to negatively impact both human and animal population. In this regard, there is an increasing emphasis on the use of alternative products, which possess distinguished advantages such as availability, safety, multiple beneficial properties. Comprehensive research providing the scientific knowledge required for the development of natural sources based pharmaceutical products represents worldwide a priority [34]. The genus *Thymus* belongs to Lamiaceae family and includes a considerable number of species and varieties of wild growing plants, with important antimicrobial activities [19,25,35,36]. Numerous researchers investigated and reported complex antimicrobial activity of *Thymus* species suggesting therapeutic potential and applications in both human and veterinary medicine [3,8,34,36,37,38,39,40,41,42]. Several derived samples widely used in folk medicine, both herbal extracts and essential oils, were previously screened and demonstrated relevant antimicrobial, antifungal and antiseptic properties [(8,34–36,38–40,42)]. Nevertheless, the broad majority of these studies were aimed to evaluate products obtained from *Thymus vulgaris* L., one of the most important species of the genus [(19,39,41,43)] and less data are documented by the literature regarding other species [8,43,44,45]. In addition, discrepancies are noticed when comparing similar studies results concerning the antimicrobial spectrum (Gram positive and/or Gram-negative bacteria, reference versus clinical strains) [34,37,45]. Also, *Thymus* essential oils are generally regarded more active against bacteria compared to other types of extracts (aqueous and alcoholic) [8,35,36]. The general consensus is that the chemical composition of the product is responsible for such quantitative and qualitative variation of the biological effect. This aspect needs to be addressed in case of *Thymus* genus in relation to published reports underlining marked chemical differences between certain species or varieties [8,19,35,36,37,43].

Few data are available in literature about *T. marschallianus*. The antimicrobial activity of the essential oil obtained from *T. marschallianus* growing in the wild in Xinjiang was demonstrated against *Escherichia coli*, *Staphylococcus aureus*, *Bacillus subtilis*, *Saccharomyces cerevisiae*, *Rhizopus sp.*, and *Penicillium sp.*, with inhibition zones and MIC values between 5.0 to 35.7 mm in diameter and 1.81 to 4.52 µL/mL, respectively [44]. In our study, six bacterial strains (*Staphylococcus aureus* ATCC 25923, *Staphylococcus pseudintermedius* ATCC 49444, *Bacillus cereus* ATCC 14579, *Enterococcus faecalis* ATCC 29219, *Salmonella typhimurium* ATCC 14,028 and *Salmonella enteritidis* ATCC 13076) were considered for the in vitro evaluation of the tested extracts antibacterial potential. All these six reference strains are relevant for this assessment given their medical relevance for both human and veterinary field. Both clinical and environmental strains of *Staphylococcus aureus*, *Staphylococcus pseudintermedius*, *Salmonella enteritidis*, *Salmonella typhimurium*, and *Enterococcus faecalis* are currently regarded as prototypes for the emergence of the antimicrobial resistance, while *Bacillus cereus* is one of the most important agents related to foodborne diseases. Moreover, this selection of strains was considered appropriate for the One Health approach, currently a widely promoted concept used for a more comprehensive understanding of health-related aspects. [46].

## 3. Materials and Methods

### 3.1. Chemicals and Instrumentation

Acetic acid, acetonitrile, methanol, gallic acid, rosmarinic acid, rutin, hydrochloric acid, sodium acetate, quercetin, Folin–Ciocâlteu reagent, protocatequic acid, gentisic acid, chlorogenic acid, catechin, caffeic acid, elagic acid, sinapic acid, ferulic acid, myricetin, tilirosid, quercetin, trans-cinnamic acid, luteolin were purchased from Sigma-Aldrich (Steinheim, Germany). Aluminum chloride, sodium acetate, sodium carbonate, ethanol was purchased from Merck, Darmstadt, Germany and DPPH (2,2-diphenyl-1-picrylhydrazyl) and BHT (butylated hydroxytoluene) were obtained from Alfa-Aesar (Germany). All microorganism strains were distributed by Oxoid^®^. All spectrophotometric data were acquired using a Jasco V-530 UV-Vis spectrophotometer (Jasco International Co., Ltd., Tokyo, Japan).

### 3.2. Plant Material and Extraction Procedure

The blooming aerial parts of *Thymus marschallianus* were harvested in June 2018, from the spontaneous flora of North Eastern Moldavia Flora, Cricova surroundings (voucher No. 959) and the culture of this species was initiated in the Experimental Fields of the Botanical Garden of the Moldavian Science Academy (voucher No. 960). The identity of the species was established by Nina Ciocârlan, PhD and the voucher specimens were deposited in the Herbarium of the Botanical Garden of the Moldavian Science Academy. Morphological characters regarding trichomes were given special attention in the identification process, in order to remove doubts regarding the possible substitution with the related species, *T. pannonicus.* Therefore, before the analysis, all tested samples were studied in order to infirm the presence of long multi-cellular trichomes and it has been established that all samples included in the study have glabrous leaves. Thus, identity of samples as *T. marschallianus* has been confirmed. The vegetal material was grinded to fine powder (300 µm), after air drying at room temperature in shade. Extract preparation was made by maceration with 70% ethanol for 10 days, at room temperature and the ratio between the vegetal material and solvent was 2/10 (g/mL). The samples were cooled down centrifuged at 4500 rpm for 15 min, and the supernatant was recovered [32,47,48].

### 3.3. HPLC-DAD-ESI (+) MS

#### Apparatus and Chromatographic Conditions

Analysis of polyphenols was performed using an Agilent Technologies 1200 Series HPLC chromatograph (Chelmsford, MA, USA), with a G1311A quaternary pump, G1322A degasser, G1329A autosampler, coupled with DAD detector and MS single quadrupole Agilent 6110 detector (Agilent Technologies, CA, USA). Separation of compounds was carried out on an Eclipse XDB-C18 column (5 μm, 4.6 × 150 mm, Agilent Technologies), using as mobile phase a mixture of 0.1% (*v*/*v*) acetic acid in distilled water and 0.1% (*v*/*v*) acetic acid in acetonitrile, at a flow rate of 0.5 mL/min. Elution was performed in gradient mode, using the following gradient: 0–2 min, 5% B; 2-18 min, 40% B; 18–20 min, 90% B; 20–24 min, 90%; min 24–25, 5% B; min 25–30, 5% B. 0–30 min, for 30 min, at a temperature of 25 °C. All samples were filtered through a 0.45-μM membrane filter (Millipore, USA) and all solvents were of HPLC grade. Detection of compounds was performed by DAD and the absorbance spectra were collected continuously in the course of each run. Identification of polyphenolic compounds was carried out by comparing the retention times, UV-visible and mass spectra of unknown peaks with the reference standards and using the database found at the following link: http://phenol-explorer.eu/compounds. The MS data were obtained using ESI positive mode. The ionization was performed in following conditions: capillary voltage 3000 V, temperature 300 °C, nitrogen flow rate 8 L/min, with scanning range between 100 and 1000 m/z, full-scan. The chromatograms were registered at 340 nm. Quantitative determinations were performed using the external standard method. The phenolic acids were expressed as mg chlorogenic acid/g d.w and flavonoids were expressed in mg rutin acid/g d.w. The calibration curves for the standards were strictly linear (R^2^ > 0.999) in the concentration range of 10-1500 μg/mL. Results were expressed as mg compound/g d.w. The chromatographic data were processed using ChemStation and DataAnalysis software from Agilent (Rev B.04.02 SPI, Palo Alto, CA, USA) [20,21,22].

### 3.4. Quantification of Total Polyphenols, Flavonoids, and Phenolic acids Content

The total phenolic content (TPC) was spectrophotometrically determined by a method using the Folin–Ciocalteau reagent, according to the European Pharmacopoeia. 2.0 mL of each ethanolic extract were mixed with 1.0 mL of Folin–Ciocalteu reagent, 10.0 mL of distilled water and the mixture was diluted to 25.0 mL with a 290 g/L solution of sodium carbonate. The absorbance was measured at 760 nm after 30 min. The calibration curve was made using as reference a calibration curve plotted using gallic acid. Gallic acid concentrations that were used were set at 0.02, 0.04, 0.06, 0.08, and 0.10 mg/mL and prepared in a mixture of methanol and water (50:50, *v*/*v*). TPC values were calculated using the calibration curve of gallic acid graph (R^2^ = 0.9953) [49,50,51].

The quantitative determination of flavonoids (TFC) was performed by the spectrophotometric method using aluminum chloride. 5.0 mL of each extract were mixed with 5.0 mL of sodium acetate 100 g/L, 3.0 mL of aluminum chloride 25 g/L, and diluted to 25 mL by methanol in a calibrated flask. The absorbance was measured at 430 nm. Total flavonoids content values were determined using an equation obtained from calibration curve of the rutin graph (R^2^ = 0.9996) [49,52].

The quantitative determination of phenolic acids was analyzed spectrophotometrically, in a method according to the 10^th^ Edition of the Romanian Pharmacopoeia (*Cynarae folium* monograf), using Arnows reagent (10.0 g sodium nitrite and 10.0 sodium molybdate in 100 mL distilled water) and the results were expressed as mg of rosmarinic acid equivalents (RAE)/g of dry vegetal material, calculated using a rosmarinic acid calibration curve graph (R^2^ = 0.9985). All experiments were performed in triplicate [49,50].

### 3.5. Antibacterial Activity Test

The in vitro antimicrobial potential of ethanolic extracts was evaluated by well-diffusion method against the following reference strains: *Staphylococcus aureus* ATCC 25923, *Staphylococcus pseudintermedius* ATCC 49444, *Bacillus cereus* ATCC 14579, *Enterococcus faecalis* ATCC 29219, *Salmonella typhimurium* ATCC 14,028 and *Salmonella enteritidis* ATCC 13076. After incubation of these strains with samples, all plates were examined for the determination of their zones of growth inhibition and diameters of these zones were measured in millimeters. For each bacterium, an inoculum was prepared by suspending 24 h pure culture in Mueller Hinton (MH) broth in order to dilute approximately to 10E6 colony forming unit (CFU)/mL according to McFarland scale. The bacterial suspensions were “flood-inoculated” onto the surface of MH agar and dried. The extracts (60 μL) were placed into wells (three wells of six-millimeter diameter for each extract) made into the MH agar using a sterile cork-borer. The assay included 70% ethanol as the negative control and gentamicin as standard antibiotic. All tests were performed in triplicate. The diameters of the growth inhibition zones (Appendix A) were measured after 24 h of incubation at 37 °C [53].

The MICs and the MBCs of the extracts were determined by a broth microdilution method (Appendix A) [(53)]. Two-fold serial dilutions of each extract were performed adding 100 µL nutrient broth in a 96-well plate and 100 µL of plant extract in the first lines, respectively; 100 µL of initial dilution were aspirated and added to the second well line of the plate. This step was repeated to obtain the following dilutions: 50.0 µL, 25.0, 12.5, 6.25, 3.12, 1.56, 0.78, and 0.39 µL of plant extracts in 100 µL broth; 5.0 µL of a 24 h bacterial inoculum prepared as described for the well-diffusion method were placed in each well and further incubated for 24 h at 37°C, when the MICs values were read as the lowest concentrations able to inhibit the visible growth of bacteria (no turbidity in the well), when compared to the negative control (broth). After MICs values were read, 10.0 µL from each well were transferred on Mueller Hinton (MH) agar plates and incubated for 24 h at 37 °C. The lowest concentrations associated with no visible bacterial growth on the agar plates (no colonies) were read as the MBCs values.

The MIC index was also calculated for each extract based on the ratio MBC/MIC to evaluate whether the extract exhibits bactericidal (MBC/MIC < 4) or bacteriostatic (MBC/MIC > 4) effect against the tested bacterial strains [54].

### 3.6. Statistical Analysis

All the samples were analyzed in triplicate. The average and the relative SD were calculated using the Excel software package. The differences between the chemical profile and the antimicrobial activity of the two extracts were evaluated using one-way analysis of variance (ANOVA), considering significant *p* < 0.05. In addition, CORREL function was used to calculate Pearson’s correlation coefficients for the analyzed data.

## 4. Conclusions

*Thymus marschallianus* is a species that has created a lot of taxonomy confusions among botanists and, as its separate identity becomes clearer, it appears important to study its chemical composition and biological activities. In order to supply new information on *T. marschallianus*, its polyphenolic composition and the antimicrobial activities were evaluated, both for samples obtained from spontaneous flora and from culture. Polyphenols analysis allowed the identification and quantification of valuable bioactive components such as rosmarinic acid and of its methylated derivative, together with luteolin, quercetin, apigenin, and their glycosides. The present study represents the first report in scientific literature for the presence of these compounds. Among these polyphenolic compounds, the flavonoids appear to be found in the highest amounts in the sample harvested from spontaneous flora. Additionally, both samples displayed promising antibacterial potential particularly towards Gram positive organisms. Results of this study suggest the great value of this species, both from culture and spontaneous flora, to obtain phytomedicines with antimicrobial activities. They also represent a starting point towards establishing the chemical profile of this species, which can contribute to avoiding future confusion about the species of the *Thymus* genus.

## Figures and Tables

**Figure 1 molecules-24-03101-f001:**
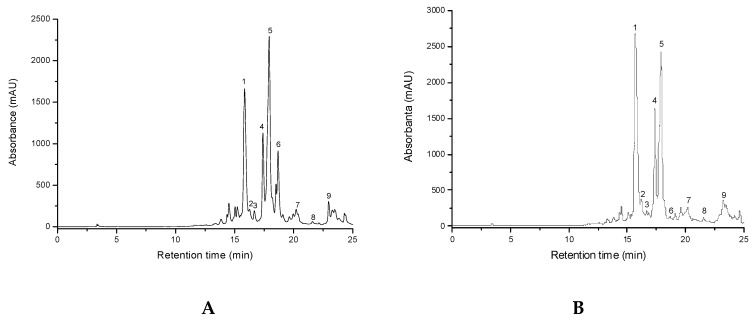
HPLC chromatograms for *T. marschallianus* from culture (TMc) (**A**) and *T. marschallianus* (TMs) (**B**). *Notes*: Chromatographic conditions were as given in the Experimental section. The identified compounds: 1. luteolin-7-*O*-glucuronide, 2. quercetin-3-*O*-glucoside, 3. quercetin-7-*O*-arabinoside, 4. apigenin-7-*O*-glucuronide, 5. methyl-rosmarinate, 6. rosmarinic acid, 7. quercetin, 8. luteolin, 9. apigenin.

**Table 1 molecules-24-03101-t001:** HPLC quantification of polyphenolic compounds from *T. marschallianus.*

Polyphenolic Compound/no.peak	[M − H] *m*/*z*	R_t_ ± SD (min)	Concentration (mg/g d.w.)	
			TMc	TMs
Luteolin-7-*O*-glucuronide/1	463	15.69 ± 0.07	20.20 ± 0.85	38.63 ± 0.92 **
Quercetin-3-*O*-glucoside (isoquercitrin)/2	465	16.20 ± 0.04	2.56 ± 0.05	4.11 ± 0.12 **
Quercetin-7-*O*-arabinoside/3	435	16.65 ± 0.05	2.09 ± 0.09 **	1.64 ± 0.07
Apigenin-7-*O*-glucuronide/4	447	17.35 ± 0.08	10.32 ± 0.72	17.02 ± 0.89 **
Methyl-rosmarinate/5	375	17.90 ± 0.10	29.39 ± 0.97 *	24.68 ± 0.85
Rosmarinic acid/6	361	18.68 ± 0.10	8.44 ± 0.73 **	1.40 ± 0.02
Quercetin/7	303	20.18 ± 0.13	2.06 ± 0.07	4.64 ± 0.12 **
Luteolin/8	287	21.55 ± 0.08	0.84 ± 0.03	2.36 ± 0.09 **
Apigenin/9	271	23.23 ± 0.06	2.77 ± 0.03	4.87 ± 0.12 **

Notes: Values represent the mean ± standard deviations, SD (*n* = 3), ** *p* < 0.001, * *p* < 0.01.

**Table 2 molecules-24-03101-t002:** The content of total polyphenols, flavonoids, and phenolic acids for *T. marschallianus* extracts.

Samples	TPC (mg GAE/g d.w.)	TFC (mg RE/g d.w.)	TPA (mg RAE/g d.w.)
**TMc**	61.99 ± 0.31	16.69 ± 0.51	26.51 ± 0.61
**TMs**	59.89 ± 0.42	28.98 ± 0.32	25.48 ± 0.23

Note: Each value represents the mean ± standard deviations of three independent measurements. GAE: Gallic acid equivalents; RE: Rutin equivalents, RAE: Rosmarinic acid equivalents. Values represent the mean ± standard deviations, SD (*n* = 3).

**Table 3 molecules-24-03101-t003:** Results of the antimicrobial preliminary screening using the well-diffusion method.

Zone of Inhibition (mm)						
Samples	*Staphylococcus aureus*	*Staphylococcus pseudintermedius*	*Bacillus cereus*	*Salmonella enteritidis*	*Salmonella typhimurium*	*Enterococcus faecalis*
**TMc**	17.33 ± 0.58	16.67 ± 0.58	16 ± 0.0	10 ± 0.0	n.a.	n.a.
**TMs**	22.67 ± 0.58	19.33 ± 0.58	20.33 ± 0.58	10.33 ± 0.58	n.a.	10 ± 0.0
**Gentamicin**	18 ± 0.00	16 ± 0.00	21 ± 0.00	18 ± 0.00	17 ± 0.00	17 ± 0.00

Notes: Values represent the mean ± standard deviations (*n* = 3); n.a.= not active.

**Table 4 molecules-24-03101-t004:** Minimum inhibitory concentration (MIC) index calculated as the ratio of the MICs and Minimum bactericidal concentrations (MBCs) values.

MIC index MBC (µg/mL)/MIC (µg/mL)						
Samples	*Staphylococcus aureus*	*Staphylococcus pseudintermedius*	*Bacillus cereus*	*Salmonella enteritidis*	*Salmonella typhimurium*	*Enterococcus faecalis*
**TMc**	8.01 (625/78)	2 (625/312)	1 (39/39)	16.02 (625/39)	n.a.	n.a.
**TMs**	8 (312/39)	1 (312/312)	1 (39/39)	1 (625/625)	n.a	1 (39/39)

Notes: Values represent the mean ± standard deviations (*n* = 3); n.a.= not active.

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
