# Peer review of "Phytochemical Profile and Antimicrobial Potential of Extracts Obtained from Thymus marschallianus Willd"

_molecules, 2019, doi:10.3390/molecules24173101_

Round 1
Reviewer 1 Report
The manuscript submitted by Niculae et al. analyzed the HPLC profiles T. marschallianus Willd. harvested from the spontaneous flora (TMs) and from culture (TMc) and compared their antimicrobial activities. The following are comments that require the authors to address:
Major Points
It is not clear about how the secondary metabolite contents and activity of the same plant spp obtained from different environments can help in addressing the taxonomical complexities associated with the plant. The authors should clarify the link between the plant contents and its taxonomy and how accurate this method is compared to genetic analysis approach. Table 3:(1) For TMc, “16,67” should read “16.67”
(2) Why are all the standard deviations for the zone of inhibition in gentamicin treatment 0.00?
It would be better to show the original pictures of antibacterial activity test as supplementary data. Line 200: “Values obtained for MICs and MBCs (Table 5) indicated a higher antimicrobial efficacy of the……”. Where is Table 5? The authors tested the antimicrobial activities of the extracts of TMc and TMs. Since there have been nine compounds detected in marschallianus (Table 1), why didn’t the authors test the antimicrobial activities of the individual compounds listed in Table 1? Similarly, Line 351: “These polyphenolic compounds are responsible for the antimicrobial properties of the extracts” is an overstatement and needs experimental evidence. The authors should perform antibacterial assay with these compounds at the concentrations detected in the TMc and TMs to support their claim. Also, it appears that TMc has higher phenolic content than TMs and yet, TMs demonstrated better antimicrobial activity (Tables 1 and 2). This could mean than the polyphenolic contents described are not the only secondary metabolites responsible for the antimicrobial activity. The authors used various descriptive words such as “superior” and “higher” to indicate difference between TMc and TMs. However, for the most part there appear to be no major differences between TMc and TMs in both content and antimicrobial attributes. The authors should perform statistical analysis if they deem the difference important. What is the rationale for testing against Staphylococcus aureus, Staphylococcus pseudintermedius, Bacillus cereus, Salmonella enteritidis, Salmonella typhimurium, and Enterococcus faecalis in antibacterial activity test? The authors should explain in their manuscript. The authors indicated in the abstract (lines 25-28) that the samples derived from culture (TMc) exhibited both greater flavonoid and rosmarinic acid contents and better antibacterial activity than those obtained from the spontaneous flora (TMs). However, in the conclusion, the authors indicated the reverse (lines 351-353). This should be addressed. Also, it would be better to include the statistical analysis to highlight the difference.
Minor Points
Line 336-338:The MIC index was also calculated for each extract based on the ratio MBC/MIC to evaluate whether the extract exhibits bactericidal (MBC/MIC < 4) or bacteriostatic (MBC/MIC > 4) effect against the tested bacterial strains. What if MBC/MIC=4? Is it bactericidal or bacteriostatic?
Line 248: ‘June’ Line 299: ‘diluted’ Line 311: ‘following’Author Response
Dear Reviewer 1,
Please see the attachment.
Best regards

Reviewer 2 Report
The present manuscript deals with the chemical composition and in vitro antimicrobial activity of hydroalcoholic extract of wild and cultivated Thymus marschallianus Willd. The methods applied are up-to-date an appropriate, the conclusions are supported by the results. The data are of interest. However, there are some points which need additional attention by the Authors:
In the Introduction, the part concerning the taxonomic problem Thymus marschallianus/T. pannonicus All. is too long. This problem is not the focus of the study; thus this part should be shortened. The identification of compounds by HPLC-DAD-MS is not described. How qas it performed? What compounds were used for calibration in the quantification procedure? The Authors should discuss their data in comparison with the published data on individual phenolics in T. marschallianus [CHENG, J., HE, K., CHEN, Q., & LIU, C. (2016). Chinese Journal of Pharmaceutical Analysis, 36(2), 285-290. Abstract available at:https://www.ingentaconnect.com/content/jpa/cjpa/2016/00000036/00000002/art00015] and T. pannonicus [Ćebović, T., Arsenijević, J., Drobac, M., Živković, J., Šoštarić, I., & Maksimović, Z. (2018). Potential use of deodorised water extracts: polyphenol-rich extract of Thymus pannonicus All. as a chemopreventive agent. Journal of food science and technology, 55(2), 560-567] In Table 2, the mean values are represented with three digits after the decimal point. However, the SDs are represented with two digits after the decimal point. In such cases the third digit after the decimal point is meaninglessAuthor Response
Dear Reviewer 2,
Please see the attachment.
Best regards.

Round 2
Reviewer 1 Report
The authors addressed most of the questions that were raised in the first round of review of their article. Although generally improved, there are still some minor points that the authors should address in their manuscript.
I still maintain that mentioning the aim/ purpose of the study three times in the last paragraph of the introduction is redundant and unnecessary. The authors should summarize the aims/purpose and condense the paragraph. Line 87; change “all these in order” to “as means to” Line 111; please change “majoritary” to “majority”. Line 135; “represent a starting point for more detailed studies that can help in identifying the chemical composition of the two…” Line 236, change “majoritary” to “majority”.Author Response
Dear Reviewer 1,
Point 1: The authors addressed most of the questions that were raised in the first round of review of their article. Although generally improved, there are still some minor points that the authors should address in their manuscript.
I still maintain that mentioning the aim/ purpose of the study three times in the last paragraph of the introduction is redundant and unnecessary. The authors should summarize the aims/purpose and condense the paragraph.
Response 1: The suggested changes to the aim/ purpose of the study were performed. The corresponding paragraph was condensed.
Point 2: Line 87; change “all these in order” to “as means to” Line 111; please change “majoritary” to “majority”. Line 135; “represent a starting point for more detailed studies that can help in identifying the chemical composition of the two…” Line 236, change “majoritary” to “majority”.
Response 2: The suggested changes have been performed.
Changes are highlighted in yellow in the manuscript and in the present letter are addressed in red in order to be easily visible.
I thank you for the suggestions that helped us to provide an improved form of the manuscript.
Sincerely yours,
Mihaela Niculae, PhD
Reviewer 2 Report
Tha Authors vave given appropriate answers to all questions.
Author Response
Dear Reviewer 2,
The suggested changes regarding the English language were performed.
Changes are highlighted in yellow in the manuscript and in the present letter are addressed in red in order to be easily visible.
I thank you for the suggestions that helped us to provide an improved form of the manuscript.
Sincerely yours,
Mihaela Niculae, PhD